# Immunogenicity and Protective Efficacy of Aerosolized Live-Attenuated Yellow Fever 17D Vaccine in Mice

**DOI:** 10.3390/vaccines12080856

**Published:** 2024-07-30

**Authors:** Feng Zhu, Meng-Xu Sun, Suo-Qun Zhao, Cheng-Feng Qin, Jin-Hua Wang, Yong-Qiang Deng

**Affiliations:** 1School of Life Sciences, Southwest Forestry University, Kunming 650224, China; m15237600075@163.com; 2State Key Laboratory of Pathogen and Biosecurity, Beijing Institute of Microbiology and Epidemiology, AMMS, Beijing 100071, China; sunnyjoy@outlook.com (M.-X.S.); suoqun_zhao@163.com (S.-Q.Z.); qincf@bmi.ac.cn (C.-F.Q.)

**Keywords:** yellow fever virus, aerosol, immunization route, immunogenicity, protective efficacy

## Abstract

Yellow fever (YF), caused by the yellow fever virus (YFV), continually spreads and causes epidemics worldwide, posing a great threat to human health. The live-attenuated YF 17D vaccine (YF-17D) has been licensed for preventing YFV infection and administrated via the intramuscular (i.m.) route. In this study, we sought to determine the immunogenicity and protective efficacy of aerosolized YF-17D via the intratracheal (i.t.) route in mice. YF-17D stocks in liquids were successfully aerosolized into particles of 6 μm. Further in vitro phenotype results showed the aerosolization process did not abolish the infectivity of YF-17D. Meanwhile, a single i.t. immunization with aerosolized YF-17D induced robust humoral and cellular immune responses in A129 mice, which is comparable to that received i.p. immunization. Notably, the aerosolized YF-17D also triggered specific secretory IgA (SIgA) production in bronchoalveolar lavage. Additionally, all immunized animals survived a lethal dose of YFV challenge in mice. In conclusion, our results support further development of aerosolized YF-17D in the future.

## 1. Introduction

Yellow fever virus (YFV), a prototype of the orthoflavivirus, is an enveloped virus with a single-stranded, positive-sense RNA genome of about 11,000 nucleotides. A single open reading frame encodes a 3400 amino acid polyprotein that is cleaved into ten viral proteins: three structural (Capsid, prM, and E) and seven nonstructural (NS1, NS2A, NS2B, NS3, NS4A, NS4B, and NS5) [1,2]. YFV is transmitted from human to human by the infected Aedes aegypti mosquitoes [3]; YFV infection can cause a range of symptoms, from asymptomatic cases to death. Severe cases may exhibit high fever, multiple organ dysfunction (liver, kidneys, heart), or hemorrhage [4,5,6]. At present, YFV continues to cause periodic large outbreaks in Africa and South America, and there are approximately 200,000 YFV cases with 30,000 fatalities worldwide annually [2]. Especially, 4347 suspected cases and 377 suspected deaths were recorded during the 2016 Angola epidemic [1], and 2251 cases and 772 deaths in Brazil [7]. Currently, there is no specific antiviral treatment against YFV infection. Control measures mainly rely on the administration of a live-attenuated YF vaccine (YF-17D) [8].

YF-17D is a highly effective, well-tolerated, live-attenuated vaccine, with more than one billion doses administered worldwide over the past 80 years [9]. Previous studies have indicated that YFV-specific antibodies can be detected in the serum within 10 days of vaccination, followed by the production of high levels of neutralizing antibodies and a large number of memory T-cells [8]. Another study has shown that more than 70% of vaccine recipients still possess protective antibodies against YFV even after more than 10 years of vaccination [10]. Furthermore, the study has also shown that a single dose of YF-17D can provide persistent immunity against YFV for up to 35 years, with detectable antibodies for up to 40 years [11,12]. Especially, the YF-17D can be used as a viral vector to develop chimeric vaccines against other important viruses, such as Dengue virus, West Nile virus, Japanese encephalitis virus, tick-borne encephalitis virus, Zika virus, Ebola virus, and SARS-CoV-2 [1,13,14].

Currently, YF-17D is primarily administered through subcutaneous (s.c.) and intramuscular (i.m.) routes and generally requires cold chain transportation [1,15,16]. Additionally, s.c., i.m., and the intraperitoneal (i.p.) route are commonly used in animal models to access the infection or immunization of YF-17D or YF-17D based vaccines [8,17,18]. The intraperitoneal route is often used in YF-17D-related mouse immunization experiments to achieve the greatest degree of antigen exposure and consistency in vaccination outcomes [8,14,17,19]. Accordingly, despite the enormous success of the YF-17D vaccine in preventing yellow fever, better vaccination strategies still need to be explored in order to guarantee a protective immune response, fewer side effects, and a low cost for all populations. Compared to vaccines administered through the s.c. and i.m. routes, mucosal vaccination can not only induce systemic immune responses, but also trigger local IgA responses in the mucosa, which is emerging as a novel vaccination strategy [20].

At present, mucosal tissues are predominantly found in the skin, respiratory tract, gastrointestinal tract, and reproductive tract. In particular, due to the presence of numerous alveolar macrophages and dendritic cells in the lungs, as well as the proximity of the bronchus to relevant lymphoid tissue, the pulmonary delivery of vaccines offers several advantages compared to traditional vaccination methods [21]. For instance, the large surface area of the respiratory system facilitates vaccine absorption; the non-invasive nature of aerosol delivery reduces the risk of spreading blood-borne diseases; the simplicity of respiratory delivery allows for self-administration, reducing the burdens on healthcare professionals [15,20,22]. In fact, previous studies also have demonstrated that a mucosal vaccination strategy can induce long-lasting systemic and mucosal immune responses against influenza [23], measles virus, and botulinum neurotoxin A [15,24]. However, the immunogenicity and protective efficacy of aerosolized 17D by the intratracheal (i.t.) route still remains unknown.

In this study, we evaluate the immunogenicity of aerosolized YF-17D by the i.t. route in A129 mice compared to the i.p. route, including IgG, IgA, and neutralizing antibodies response in sera, SIgA production in the bronchoalveolar lavage, as well as cellular immunity in the spleen. Meanwhile, we also determined the protective efficacy of aerosolized YF-17D in mice after the intracranial (i.c.) YFV challenge. Our results provide a scientific basis for the rational immunization of aerosolized YF-17D in the future.

## 2. Material and Methods

### 2.1. Cells and Viruses

BHK-21 cells (#CCL-10) were obtained from ATCC and cultured in Dulbecco’s modified Eagle’s medium (DMEM) with 10% fetal bovine serum (FBS), 100 U/mL penicillin and 100 g/mL streptomycin. Aedes albopictus C6/36 cells (CRL-1660) were grown in RPMI 1640 medium (Gibco, Carlsbad, CA, USA) with 10% FBS at 30 °C. The 17D virus (GenBank accession no. MN072725) was recovered from the pYF-17D plasmid. Viral stocks were prepared on a monolayer of C6/36 cells, titrated in BHK-21 cell by plaque-forming assay, and stored at −80 °C.

### 2.2. Aerosolization of YF-17D

YF-17D solution was aerosolized using a hand-held liquid aerosol pulmonary delivery device (HLAPDD) (TOW TECH, Shanghai, China) for mice. This device generates aerosols that are delivered directly into the lungs for precise quantification. Aerosol particle size was measured with an aerodynamic particle sizer (APS 3321, TSI, Shoreview, MN, USA) at a flow rate of 5 L/min for 15 s. The mean mass aerodynamic diameter (MMAD) was determined using the Aerosol Instrument Manager Software (8.1.1.0).

### 2.3. In Vitro Phenotypic Characteristics of Aerosolized YF-17D

The 17D aerosol was collected in a 1.5-mL EP tube and centrifuged for 1 min at 8000 rpm for quantitative reverse transcription PCR (RT-qPCR), immunofluorescence staining, and plaque assays. Virus titer, replication curve, and protein expression were compared with the original solution.

For virus titers, BHK-21 cells were infected with 10-fold dilutions of the virus for 1 h at 37 °C. The supernatants were replaced with DMEM containing 1% low melting point agarose (Promega, Madison, WI, USA) and 2% FBS. Four days post-infection, the cells were fixed with 4% formaldehyde and stained with 1% crystal violet. Visible plaques were counted, and the final titers were calculated as plaque-forming units per mL (PFU/mL).

For the replication curve, BHK-21 cells were cultured in 24-well plates at 37 °C with 5% CO_2_. Liquid and aerosolized YF-17D were added and incubated for 1 h at 37 °C. Infected cells were then cultured at 37 °C with 5% CO_2_, and supernatants were collected at 24, 48, and 72 h post-infection. The viral RNA load in the supernatant was measured by RT-qPCR, and replication curves were plotted using GraphPad Prism 8 software.

To analyze viral protein expression, infected BHK-21 cells were fixed with methanol/acetone (7:3) at 48 h post-infection. Cells were blocked for 1 h with 1% BSA in PBS at 37 °C, then incubated with the orthoflavivirus group E protein monoclonal antibody (1:1000, GeneTex, GTX57154, Irvine, CA, USA) at 37 °C for 2 h. Subsequently, goat anti-mouse IgG-Alexa Fluor 488 (1:1000, Proteintech, Rosemont, IL, USA) was added and the plates incubated at 37 °C for 1 h. Cell nuclei were stained with DAPI (Solarbio, Beijing, China) for 5 min. Fluorescence images were captured using a Zeiss Axio Observer inverted microscope with a 10× objective.

### 2.4. Animal Experiments

To evaluate the immunogenicity of aerosolized YF-17D in IFN-α/β receptor deficient (IFNAR^−/−^) mice (A129), 6~8-week-old A129 mice were obtained from the Laboratory Animal Center (AMMS) and inoculated with 10^4^ PFU of YF-17D by the i.p. or i.t. routes. At 14 and 21 dpi, sera samples were collected to determine total YFV-specific IgG and IgA antibody titers by Enzyme-linked immunosorbent assay (ELISA), and neutralizing antibody (nAb) titers by plaque reduction neutralization test assay (PRNT), respectively.

To evaluate the cellular immune response and SIgA antibody levels induced by aerosolized YF-17D in A129 mice, groups of A129 mice were inoculated with 10^4^ PFU of YF-17D via the i.t. and i.p. routes. At 21 dpi, immunized mice were euthanized, and spleens were harvested to determine T cell response by Enzyme-linked immunospot assay (ELISPOT), and bronchoalveolar lavage (BAL) fluid was collected to determine the SIgA by ELISA.

To evaluate the protective efficacy of aerosolized YF-17D in mice, immunized mice were challenged with 10^3^ PFU of YF-17D by the i.c. route at 28 dpi and monitored for clinical symptoms and mortality for 14 days. Sera samples were collected at 1, 3, and 5 days post-challenge (dpc) and indicated tissue samples (brains, hearts, livers, spleens, lungs, and kidneys) were collected at 5 dpc for the detection of YF-17D-specific RNA copies (RNA copies per ml or per gram) by RT-qPCR.

### 2.5. ELISA

IgG and IgA antibody in the sera samples and SIgA in the BAL fluid were assessed by ELISA. Briefly, 96-well plates (Costar, Corning, NY, USA) were coated overnight at 4 °C with inactivated YF-17D (10^4^ PFU/well) diluted with coating buffer (Solarbio, Beijing, China). The serum samples were heated at 56 °C for 30 min before use. Then, inactivated serum samples were three-fold serially diluted, 100µL of the diluted serum or BAL was added to the blocked plates and incubated at 37 °C for 2 h. Then, secondary antibody (HRP-Goat pAb to Mouse IgG/IgA 1:5000) (ZSGB-BIO, Beijing, China/Abcam, Cambridge, UK) or HRP-pAb to Mouse IgA) (Beyotime, Shanghai, China) was added to each well of the plates and incubated for 1 h at 37 °C. TMB single-component substrate solution (Solarbio, Beijing, China) was incubated for 15 min at room temperature. The reaction was stopped by adding stop solution (Solarbio, Beijing, China), and absorbance at 450–630 nm was measured using a Synergy H1 hybrid multimode microplate reader (BioTek, Winooski, VT, USA). Endpoint titers were defined as the highest serum dilution giving an absorbance over 2-fold above background. The result of BAL fluid was expressed as ODs measured at 450 nm. The cut-off value for positive sample was set at >2-fold of the mean value of the negative control group.

### 2.6. ELISPOT

At 21 days post-immunization, cellular immune responses were assessed by isolating splenocytes and using precoated IFN-γ and IL-4 ELISpot kits (Mabtech, Nacka Strand, Sweden) following the manufacturer’s instructions. Briefly, plates were blocked with RPMI 1640 (Thermo Fisher Scientific, Waltham, MA, USA) containing 10% FBS for 30 min. Splenocytes (1 × 10^6^ cells/well) were stimulated with YF-17D E protein peptide pools (GenScript, Nanjing, China). Concanavalin A (ConA, Sigma, St. Louis, MO, USA) was the positive control, and unstimulated cells were the negative control. After 36 h at 37 °C with 5% CO_2_, the plates were developed by adding a biotinylated detection antibody, a streptavidin-enzyme conjugate, and the substrate. Spots were counted using an AID ELISpot reader (AID, Stuttgart, Germany), and spot-forming cells (SFCs) per million cells were calculated.

### 2.7. PRNT

Neutralizing antibody (nAb) titers in serum samples were measured using a 50% plaque reduction neutralization test (PRNT_50_). Briefly, inactivated serum samples were ten-fold serially diluted, then incubated at 37 °C for 60 min with ~100 PFU of YF-17D. The samples were added to BHK-21 cell monolayers in 12-well plates and incubated at 37 °C for 60 min. The mixtures were then replaced with DMEM containing 1% low melting point agarose (Promega, Madison, WI, USA) and 2% FBS and incubated at 37 °C in 5% CO_2_ for 4 days. The cells were fixed with 4% formaldehyde and stained with 1% crystal violet. Visible plaques were counted, viral titers determined by plaque assay, and PRNT_50_ titers calculated using the Spearman-Karber method.

### 2.8. RT-qPCR

Sera and tissue homogenates were clarified by centrifugation at 8000 rpm for 10 min. Viral RNA was extracted using the Viral RNA/DNA Extraction Kits (TIANLONG, Xi’an, China) following the manufacturer’s protocol. RT-qPCR was performed with the One Step PrimeScript RT-PCR Kit (Takara, Kusatsu, Japan) using primers and probes: YF-17D-F (5′-GCACGGATGTGACAGACTGAAGA-3′), YF-17D-R (5′-CCAGGCCGAACCTGTCAT-3′), and YF-17D-P (5′-6-FAM-CGACTGTGTGGTCCGGCCCATC-BHQ1-3′). RT-qPCR was conducted on a Light-Cycler^®^ 480 Instrument (Roche Diagnostics Ltd., Basel, Switzerland). RNA copies per ml or gram were calculated from qPCR Ct values using the published method.

### 2.9. Statistical Analysis

Data were analyzed with GraphPad Prism 8.0 software. Unless specified, data are presented as mean ± SD. Analysis of variance (ANOVA) or t-test was used to determine statistical significance among different groups (* *p* < 0.05; ** *p* < 0.01; *** *p* < 0.001; **** *p* < 0.0001; n.s., not significant).

## 3. Results

### 3.1. Characterization of Aerosolized YF-17D

Firstly, the YF-17D vaccine stocks in liquid form were processed into aerosolized YF-17D using HLAPDD as previously described [25] (Figure 1A). The particle size and distribution of aerosolized YF-17D were measured using an aerodynamic particle sizer. The mean mass aerodynamic diameter (MMAD) of the resulting particles of aerosolized YF-17D was 6.31 ± 0.38 μm (Figure 1B), which was consistent with the distribution range in the lung (<10 μm). Then, the resulting aerosolized YF-17D were subjected to infectivity assay. The plaque-forming assay showed that YF-17D maintained their infectivity and formed similar plaques in BHK-21 cells after aerosolization (Figure 1C), and there was no significant difference in virus titers between pre- and after-aerosolization (Figure 1D). The RT-qPCR results further showed that YF-17D exhibited consistent replication kinetics in BHK-21 cells between pre- and after-aerosolization (Figure 1E). The immunofluorescence staining results also showed that the expression of YF-17D E protein in BHK-21 cells had no significant difference between pre- and after-aerosolization (Figure 1F). Together, these results suggested that YF-17D can be aerosolized without loss of infectivity in vitro.

### 3.2. Aerosolized YF-17D Induced Humoral and Mucosal Immune Response

To compare the humoral immune responses induced by the i.t. and. i.p. routes, we collected sera from the immunized mice at 14 and 21 days post-immunization (dpi), and determined the titers of YFV-specific IgG and IgA antibodies and neutralizing antibodies (nAb) by ELISA and PRNT assays, respectively (Figure 2A). The results showed that both routes of immunization induced similar humoral immune responses. For the i.t. and i.p. routes, IgG antibody titers in sera from immunized mice approached ~1/70,148 and ~1/40,500 at 14 dp, then ~1/101,171 and ~1/70,148 at 21 dpi, respectively (Figure 2B). NT_50_ titers approached ~1/680 and ~1/694 at 14 dpi, then ~1/609 and ~1/716 at 21 dpi, respectively (Figure 2D). Additionally, IgA antibody titers at 14 dpi also can approach ~1/245 and ~1/54, then ~1/506 and ~1/600 at 21 dpi, respectively (Figure 2C). However, there were no significant differences between the two routes of immunization.

Especially, specific SIgAs can neutralize pathogens and prevent them from invading through the mucosal epithelium [26]. Therefore, to further evaluate the mucosal immune response induced by the i.t. and. i.p. routes, we harvested the bronchoalveolar lavage (BAL) fluid from the immunized mice at 21 dpi and measured the YFV-specific SIgA level by ELISA. The results showed that no detectable YFV-specific SIgA was observed in the BAL samples of all i.p. immunized-mice; however, YFV-specific SIgA can be detected in the BAL from three of five mice immunized via i.t. route, with the average OD_450_ value of 0.235 (Figure 2E). The result demonstrated that YF-17D aerosol can induce the production of specific SIgA in BAL.

### 3.3. Aerosolized YF-17D Induced Cellular Immune Response

To compare the T cell immune response induced by the i.t. and. i.p. routes, we isolated splenocytes from the immunized mice at 21 dpi and determined the IFN-γ and interleukin (IL)-4 levels by ELISPOT analysis. The results showed that both routes of immunization induced a high number of spot-forming cells in IFN-γ, but less in IL-4 (Figure 3A). Both routes of immunization induced a comparative number of IFN-γ-releasing cells (Figure 3B), which were significantly higher than the control group (*p* < 0.05). However, levels of IL-4 secretion in both groups of immunized mice had no significant difference compared with the control group (Figure 3B). Together, these results demonstrate that aerosolized YF-17D could induce cellular responses in A129 mice comparable to the i.p. immunization.

### 3.4. Aerosolized YF-17D Conferred Full Protection against Lethal YFV Challenge

Previous studies showed that intracranial challenge of YF-17D was used to assess the protective effect of the YF-17D and YF-17D-derived vaccines against YFV [8,16,27]. Therefore, to examine the protection efficiency induced by the i.t. and. i.p. routes against lethal YFV challenges, we challenged the immunized mice with 10^3^ PFU of YF-17D via the i.c. inoculation at 28 dpi. As expected, these results showed that all the mice from the control group developed severe neurotropic disease and succumbed to death at 7 days post-challenge (dpc), while all the immunized mice survived without any clinical symptoms during the observation period (Figure 4A). Moreover, the RT-qPCR results showed that all the mice from the control group developed high levels of viremia, with average viral loads of 2.79 × 10^7^ RNA copies/mL and 6.66 × 10^8^ RNA copies/mL at 3 and 5 dpc, respectively (Figure 4B). In contrast, both routes had completely undetectable viremia at 3 dpc, and decreased by 30- and 63-fold at 5 dpc, respectively (Figure 4B). Meanwhile, three mice were anesthetized at 5 dpc, and the selected organs were measured by RT-qPCR assay. The results revealed that all the mice from the control group developed high levels of viral loads in the tested tissues. Among them, brains have the highest viral RNA loads with ~1.59 × 10^11^ RNA copies/g. Whereas both routes reduced to 5.27 × 10^9^ and 4.05 × 10^9^ RNA copies/g, respectively (Figure 4C). Additionally, viral RNA loads were barely detectable in the spleen, liver, and kidney compared to the control mice (Figure 4C). Together, these results demonstrate that the aerosolized YF-17D immunization can provide full protection against lethal YFV challenge.

## 4. Discussion

In the present study, the immunogenicity and protective efficacy of pulmonary delivery of aerosolized YF-17D was determined. Groups of A129 mice were vaccinated with 10^4^ PFU of YF-17D via i.t. and i.p. routes, respectively. Neutralizing antibodies are the major protecting factor against orthoflavivirus infection. The nAb titer in mice serum was determined via PRNT. At 14 and 21 dpi, i.t. immunization-induced nAb titers both are over 10^2.5^ that are far beyond the known surrogate correlate of protection [8,16,28]. In addition, i.t. immunization induces comparable IgG and IgA antibody titers to the i.p. immunization (Figure 2), which further demonstrates that i.t. immunization with YF-17D produces a high level of humoral immune responses.

YF-17D vaccination elicits broad and functional memory T cell responses in humans, which is an indispensable component of YF-17D-induced protective immunity [29,30]. Notably, the ELISpot analysis reveals significant induction of IFN-γ in the splenocytes from i.t immunization mice (Figure 3), which is similar to i.p. immunization and consistent with previous studies [8,16]. Together, our results indicate that i.t. immunization with YF-17D not only induces humoral immune responses but also specific cellular responses in mice.

Notably, our results reveal that i.t. immunization with YF-17D not only can induce systemic immune responses, but also trigger local IgA responses in the mucosa. Indeed, 17D-specific SIgA antibody was detected in the BAL fluid of A129 mice immunized with aerosolized YF-17D (Figure 2). Previous studies have shown that orthoflaviviruses by nasal or pulmonary delivery can induce systemic humoral immunity, but mucosal immunity has not been measured [25,31]. Our results demonstrated that lung delivery of the aerosolized YF-17D can induce the generation of local mucosal immunity in mice, which is similar to some pulmonary delivered mucosal vaccines, e.g., influenza viruses [23], measles virus [24], and SASR-CoV-2 [32]. However, more studies are needed to demonstrate whether specific mucosal immunity induced by pulmonary administration could improve protection against YFV.

As the challenge infection was performed intracranially, virus loads in the brain largely determined the fate of thus challenged mice. As expected, a high level of viral RNA load was detected in the brain of mock vaccinated animals, and all mice succumbed within 7 days post-challenge, which is consistent with previous studies [8,16]. However, i.t. vaccination dramatically restricted virus replication in the mouse brain, and significantly reduced viremia and viral loads in the spleen, liver, and kidney. Similar to the i.p. vaccination, all i.t. route-vaccinated mice survived after the intracranial challenge of YFV (Figure 4), indicating that the vaccination with YF-17D via the i.t. route can provide a full protection efficacy.

Currently, low vaccination rates are one of the main reasons for the prevalence of yellow fever in tropical regions [4,33,34]. The non-invasive nature of the mucosal immunization method does not require trained health-care personnel; meanwhile, this needle-free administration might improve the vaccinees’ compliance [15,20,22]. Therefore, aerosolized YF-17D is expected to contribute to increased YF vaccine coverage in endemic areas, especially in non-industrialized countries and remote areas.

## 5. Conclusions

In conclusion, we found that a single i.t. immunization with aerosolized YF-17D induced robust humoral and cellular immune responses in mice, and further triggered secretory IgA (SIgA) production in bronchoalveolar lavage. Moreover, the i.t. immunization of YF-17D showed good protection against intracranial challenge of YFV. In summary, our results provide an effective alternative route for YF-17D or YF-17D-based vaccines as well as other related orthoflavivirus vaccines.

## Figures and Tables

**Figure 1 vaccines-12-00856-f001:**
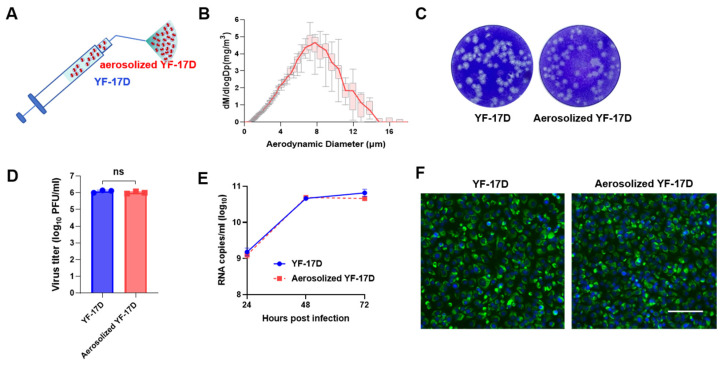
Biological characteristics of aerosolized YF-17D in vitro (**A**) Schematic of the hand-held liquid aerosol pulmonary delivery device (HLAPDD). (**B**) Aerodynamic median mass diameter (MMAD) of YF-17D aerosol measured by particle size spectrometer. (**C**) Plaque morphology of YF-17D and aerosolized YF-17D on BHK-21 cells. (**D**) The viral titers of YF-17D and aerosolized YF-17D on BHK-21 cells. (**E**) Replication curve of YF-17D and aerosolized YF-17D on BHK-21 cells. BHK-21 cells were infected with both liquid and aerosolized YF-17D. Cell supernatants were collected at 24, 48, and 72 hpi to measure the viral RNA load using the RT-qPCR assay. (**F**) Viral protein expression of YF-17D and aerosolized YF-17D on BHK-21 cells. Briefly, BHK-21 cells were infected with YF-17D and aerosolized YF-17D, then the BHK-21 cells were fixed at 48 hpi for immunofluorescence staining. The E protein of YF-17D appeared as a green fluorescence, while the cell nuclei were visualized in blue through DAPI staining. The image is referenced with a scale bar representing 100 μm. Significance was calculated using unpaired *t*-test (n.s., not significant).

**Figure 2 vaccines-12-00856-f002:**
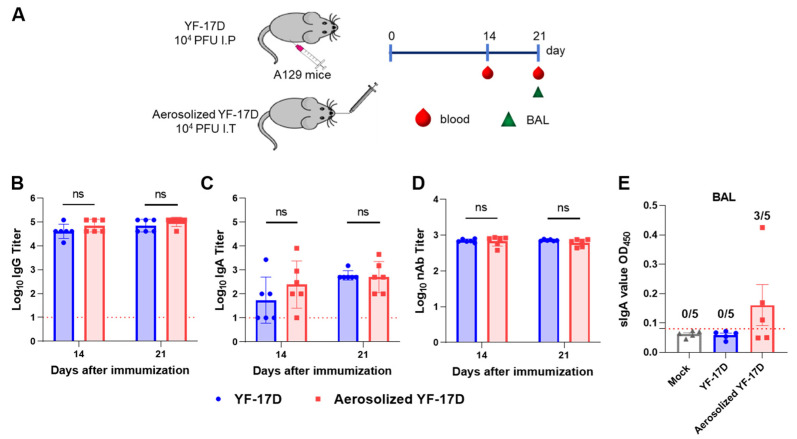
Immunogenicity of aerosolized YF-17D in A129 mice (**A**) Schematic diagram of the experimental design. Briefly, groups of 6~8-week-old A129 mice (n = 6) were immunized with 10^4^ PFU of YF-17D or aerosolized YF-17D, respectively. Then, sera samples at 14, 21 dpi were collected for total YFV-specific IgG antibody (**B**) and IgA antibody (**C**) titers by ELISA, and neutralizing antibody (nAb) (**D**) titers by PRNT, respectively. IgG and IgA antibody titers were calculated according to the highest dilution that resulted in a value two-fold greater than the absorption of the control serum. (**E**) SIgA antibody in BAL of YF-17D-immunized mice (n = 5) at 21 dpi were measured by ELISA. The positive cutoff is twice the average of the negative control group. The data were shown as mean ± SD. Significance was calculated using unpaired *t*-test (n.s., not significant).

**Figure 3 vaccines-12-00856-f003:**
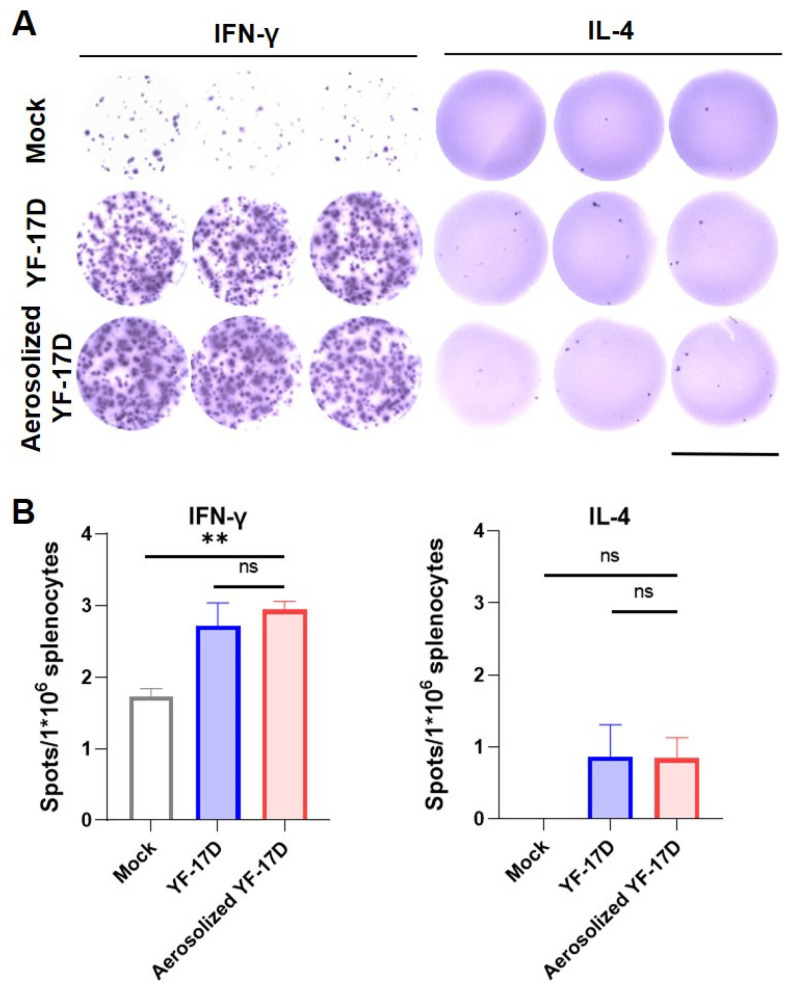
Cellular immunity of aerosolized YF-17D in A129 mice (**A**) ELISpot results measured with the AID ELISpot reader. To detect YF-17D-specific T-cell responses, cytokine screening of splenocytes was performed by IFN-γ and IL-4 ELISPOT, splenocytes harvested from A129 immunized (n = 3~4) with 1 × 10^4^ PFU of YF-17D at three weeks post-vaccination. The image is referenced with a scale bar representing 6.4 mm. (**B**) Quantification of YF-17D-specific IFN-γ and IL-4-producing T cell. The data were shown as mean ± SD. Significance was calculated using one way ANOVA (n.s., not significant; **, *p* < 0.01).

**Figure 4 vaccines-12-00856-f004:**
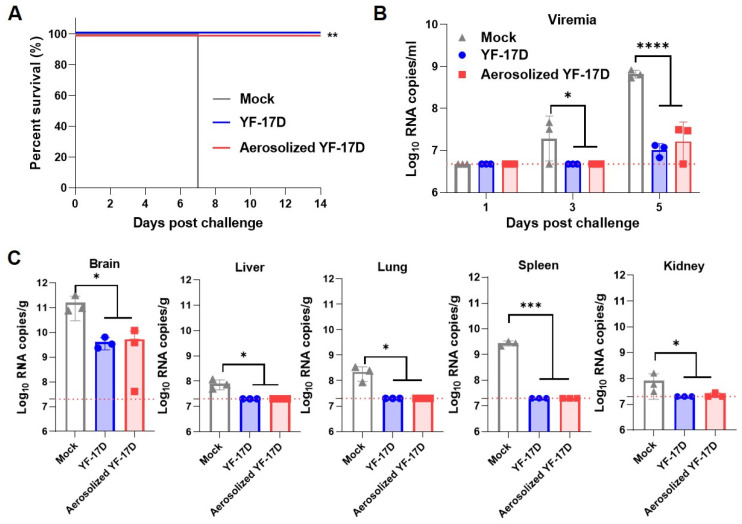
Protective efficacy of aerosolized YF-17D in A129 mice. (**A**) Survival curve of the immunized mice (n = 5). Briefly, the immunized mice were challenged with 1 × 10^3^ PFU of YF-17D via the i.c. route at 28 days after immunization and monitored for clinical symptoms and mortality. Kaplan-Meier survival curves were analyzed by a log-rank test (**, *p* < 0.01). (**B**) Viral loads in serum were measured at 1, 3, and 5 days after challenge by RT-qPCR assays. Viral load was expressed as RNA copies per milliliter. Dotted lines indicate the limit of detection. (**C**) Tissue distribution in mice (n = 3) at 5 days after YF-17D challenge. Dotted lines represent limits of detection. The data were shown as mean± SD. Significance was calculated using one way ANOVA (n.s., not significant; *, *p* < 0.05; ***, *p* < 0.001; ****, *p* < 0.0001).

## Data Availability

The data presented in this study are available on request from the corresponding author Y.Q.D.

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
