# Peer review of "Immunogenicity and Protective Efficacy of Aerosolized Live-Attenuated Yellow Fever 17D Vaccine in Mice"

_vaccines, 2024, doi:10.3390/vaccines12080856_

Round 1

Reviewer 1 Report

Comments and Suggestions for Authors

In this study, Zhu et al examined the immunogenicity and protective efficacy of aerosolized YF-17D vaccination via the intratracheal (i.t.) route in A129 mice and compared with those of liquid YF-17D vaccination via the intraperitoneal (i.p.) route. The results revealed that i.t. immunization with aerosolized YF-17D can induce humoral and cellular immune responses comparably to i.p. immunization, and trigger rather higher secretory IgA production in mucosal tissues. Moreover, the authors showed that the aerosolized YF-17D immunization completely protected mice from lethal YF-17D challenge infection accompanied with significant reductions in tissue viral loads, which are comparable to those induced by i.p. immunization. These data encourage the utility of aerosolized YF-17D especially in the area where the current YF-17D vaccines are not sufficiently supplied due to the cost issue.

Overall, the manuscript is well written. I have one comment that needs to be addressed.

YF-17D is generally administered through subcutaneous or intramuscular route in humans. However, the mice were strangely immunized with liquid YF-17D via intraperitoneal route for a comparison with the aerosolized YF-17D vaccination. In order to convince the Readers that the effect of aerosolized YF-17D is comparable to that of standard YF-17D vaccination, the authors should at least present the rationale that subcutaneous or intramuscular immunization can elicit immune responses comparably to intraperitoneal immunization.

Minor

2.2 “Aerosolization of Aerosolized YF-17D” should be “Aerosolization of Aerosolized YF-17D”.

Author Response

Reviewer #1 (Comments to the Author):
In this study, Zhu et al examined the immunogenicity and protective efficacy of aerosolized YF-17D vaccination via the intratracheal (i.t.) route in A129 mice and compared with those of liquid YF-17D vaccination via the intraperitoneal (i.p.) route. The results revealed that i.t. immunization with aerosolized YF-17D can induce humoral and cellular immune responses comparably to i.p. immunization, and trigger rather higher secretory IgA production in mucosal tissues. Moreover, the authors showed that the aerosolized YF-17D immunization completely protected mice from lethal YF-17D challenge infection accompanied with significant reductions in tissue viral loads, which are comparable to those induced by i.p. immunization. These data encourage the utility of aerosolized YF-17D especially in the area where the current YF-17D vaccines are not sufficiently supplied due to the cost issue.

Overall, the manuscript is well written. 

I have one comment that needs to be addressed.

Comment 1: YF-17D is generally administered through subcutaneous or intramuscular route in humans. However, the mice were strangely immunized with liquid YF-17D via intraperitoneal route for a comparison with the aerosolized YF-17D vaccination. In order to convince the Readers that the effect of aerosolized YF-17D is comparable to that of standard YF-17D vaccination, the authors should at least present the rationale that subcutaneous or intramuscular immunization can elicit immune responses comparably to intraperitoneal immunization. 

Response 1: Thanks for your advice. Currently, subcutaneous injection, intramuscular injection, and intraperitoneal injection are commonly used routes for infection and immunization in YF-17D-related animal experiments. (Ma J, Boudewijns R, Sanchez-Felipe L, Mishra N, Vercruysse T, Buh Kum D, et al. Comparing immunogenicity and protective efficacy of the yellow fever 17D vaccine in mice. Emerg Microbes Infect. 2021;10:2279-90; Erickson AK, Pfeiffer JK. Spectrum of disease outcomes in mice infected with YFV-17D. The Journal of general virology. 2015;96:1328-39; Piras-Douce F, Raynal F, Raquin A, Girerd-Chambaz Y, Gautheron S, Sanchez MEN, et al. Next generation live-attenuated yel-low fever vaccine candidate: Safety and immuno-efficacy in small animal models. Vaccine. 2021;39:1846-56.) However, previous study also suggested that the infectivity of YF-17D administered via intraperitoneal injection is superior to that of subcutaneous injection. (Erickson AK, Pfeiffer JK. Spectrum of disease outcomes in mice infected with YFV-17D. The Journal of general virology. 2015;96:1328-39.) Furthermore, in animal experiments related to YF-17D research, intraperitoneal injection is a frequently utilized method of immunization because it provides the greatest exposure to the antigen and ensures the uniformity of the immune response. (Ma J, Boudewijns R, Sanchez-Felipe L, Mishra N, Vercruysse T, Buh Kum D, et al. Comparing immunogenicity and protective efficacy of the yellow fever 17D vaccine in mice. Emerg Microbes Infect. 2021;10:2279-90; Sanchez-Felipe L, Vercruysse T, Sharma S, Ma J, Lemmens V, Van Looveren D, et al. A single-dose live-attenuated YF17D-vectored SARS-CoV-2 vaccine candidate. Nature. 2021;590:320-5; Erickson AK, Pfeiffer JK. Spectrum of disease outcomes in mice infected with YFV-17D. The Journal of general virology. 2015;96:1328-39; Kum DB, Mishra N, Boudewijns R, Gladwyn-Ng I, Alfano C, Ma J, et al. A yellow fever-Zika chimeric virus vaccine candi-date protects against Zika infection and congenital malformations in mice. NPJ Vaccines. 2018;3:56.) 

Therefore, we have added an explanation “Additionally, s.c., i.m., and intraperitoneal (i.p.) route are commonly used in animal models to access the infection or immunization of YF-17D or YF-17D based vaccines [8, 17, 18]. Intraperitoneal route is often used in YF-17D-related mouse immunization experiments to achieve the greatest degree of antigen exposure and consistency in vaccination outcomes [8, 14, 17, 19].” from line 64 to line 68 in the revised manuscript.

Comment 2: 2.2 “Aerosolization of Aerosolized YF-17D” should be “Aerosolization of Aerosolized YF-17D”.

Response 2: Thank you, the term 'Aerosolized' in the section title 2.2 has been removed.

Reviewer 2 Report

Comments and Suggestions for Authors

In this manuscript, Zhu et al develop a YFV 17D vaccine model in mice to examine the utility of aerosolized delivery of the vaccine to elicit humoral and cellular immunity and protect against lethal, intracranial challenge. The authors demonstrate that aerosolized YFV 17D maintains its infectious properties in vitro and that administering this aerosolized virus intratracheally elicited quantities of serum antibodies that possessed similar neutralization efficiency to those produced by intramuscular vaccination. Moreover, T-cell activation and protection from viral challenge was similar between vaccine routes. Together, these data support the use of aerosolized vaccine as a promising means to prevent YFV disease.

A major point that the authors make throughout the manuscript is the importance of capitalizing on greater induction of mucosal immunity through aerosolized immunization for better vaccine efficacy.  However, the authors do not convincingly demonstrate that (A) the induction of mucosal immunity is substantial different between the immunization routes or (B) that this is even important for vaccine effectiveness.  For point A, the authors only assessed mucosal immunity by one method (sIgA, as shown in Fig. 2E) and do not even include statistical analysis to determine whether these subtle differences are statistically significant let alone biologically significant. The authors should include statistical analysis of these data and another method to score mucosal immunity (e.g., activation of lung-resident alveolar macrophages, salivary IgG, etc) if they wish to justify and describe these findings. For part B, this minor difference in sIgA does not seem to contribute to vaccine efficacy or any correlate of protection. For both routes of immunization, other measurements of immune activation (i.e., splenic T-cell activation, serum IgG/IgA) were roughly identical and viral loads and animal survival duration were the same.   

Given the data presented herein, this reviewer thinks that the manuscript should be retailored to highlight the substantiated results of the manuscript or additional data should be included to clearly justify the authors’ conclusions. A better way to present these findings would be to highlight that these data support an alternative route of immunization of the YFV 17D vaccine to enable better delivery and implementation globally without a cost in efficacy and protection.

In addition, there are several minor edits required to improve the quality of the manuscript:

Line 26: Flavivirus genus has been renamed to Orthoflavivirus.

Line 29: Replace “Cap” with “Capsid” and “PrM “with “prM”.

Line 31: Aedes aegypti should be italicized.

Line 50: “Tick-borne” should not be capitalized.

Line 88: Delete the word “Aerosolized”.

Line 168: I think “teofold “should be “ten-fold”

Fig. 1F: Higher resolution images would help the readers see how similar/different the two virus preparations are in their infectivity.

Line 20, 74, 127, 132, 140, etc: Please be consistent with whether you use sIgA or SIgA.

Fig. 2 and Fig. 3: There is no mention of the size of groups used for the animal studies, either in the methods, figures, or figure legends.  For Fig.2B-E and Fig.3B-C represent individual animals?  Please clarify this.

Line 321: “17D-special” should be “17D-specific”

Comments on the Quality of English Language

The manuscript needs to be proofread for typographical errors and inconsistencies.  A few examples are highlighted in my comments above.

Author Response

Comment 1: A major point that the authors make throughout the manuscript is the importance of capitalizing on greater induction of mucosal immunity through aerosolized immunization for better vaccine efficacy.  However, the authors do not convincingly demonstrate that (A) the induction of mucosal immunity is substantial different between the immunization routes or (B) that this is even important for vaccine effectiveness.  For point A, the authors only assessed mucosal immunity by one method (sIgA, as shown in Fig. 2E) and do not even include statistical analysis to determine whether these subtle differences are statistically significant let alone biologically significant. The authors should include statistical analysis of these data and another method to score mucosal immunity (e.g., activation of lung-resident alveolar macrophages, salivary IgG, etc) if they wish to justify and describe these findings. For part B, this minor difference in sIgA does not seem to contribute to vaccine efficacy or any correlate of protection. For both routes of immunization, other measurements of immune activation (i.e., splenic T-cell activation, serum IgG/IgA) were roughly identical and viral loads and animal survival duration were the same. 

Response 1:Thanks for your Constructive suggestions. We have readjusted the content in the revised manuscript so that the existing data can support the relevant conclusions. Firstly, we have added an description “The results showed that no detectable YFV-specific SIgA was observed in the BAL samples of all i.p. immunized-mice, however, YFV-specific SIgA can be detected in the BAL from three of five mice immunized via i.t. route, with the average OD450 value of 0.235 (Figure 2E). The result demonstrated that YF-17D aerosol can induce the production of specific SIgA in BAL.” from line 289 to 293 in the revised manuscript. Meanwhile, we has added the positive rate of SIgA in BAL in Fig. 2E in the revised manuscript to highlight the differences in the induction of SIgA between the two immunization methods.

Comment 2:  Given the data presented herein, this reviewer thinks that the manuscript should be retailored to highlight the substantiated results of the manuscript or additional data should be included to clearly justify the authors’ conclusions. A better way to present these findings would be to highlight that these data support an alternative route of immunization of the YFV 17D vaccine to enable better delivery and implementation globally without a cost in efficacy and protection.

Response 2:  Thanks for your Constructive suggestions. We have added “Currently, low vaccination rates are one of the main reasons for the prevalence of yellow fever in tropical regions [4, 33, 34]. The non-invasive nature of mucosal immunization method does not require trained health-care personnel. Meanwhile, this needle-free administration might improve the vaccinees compliance [15, 20, 22]. Therefore, aerosolized YF-17D is expected to contribute to increased YF vaccine coverage in endemic areas, especially in non-industrialized countries and remote areas.” from line 457 to 462 in the revised manuscript.

Comment 3:  Line 26: Flavivirus genus has been renamed to Orthoflavivirus.

Response 3:  Thank you for your valuable suggestions. We have already changed "Flavivirus genus" to "Orthoflavivirus" in the revised manuscript.

Comment 4:  Line 29: Replace “Cap” with “Capsid” and “PrM “with “prM”.

Response 4:   The term has been replaced at line 28 in the revised manuscript.
Comment 5:  Line 31: Aedes aegypti should be italicized.

Response 5:  We have changed “Aedes aegypti” to “Aedes aegypti” at line 30.

Comment 6:   Line 50: “Tick-borne” should not be capitalized.

Response 6:  We have changed “Tick-borne” to “tick-borne” at line 60.

Comment 7:  Line 88: Delete the word “Aerosolized”.

Response 7:  We have detected “Aerosolized” at line 103.

Comment 8:  Line 168: I think “teofold “should be “ten-fold”

Response 8:  We have changed “teofold “ to “ten-fold” at line 212 in the revised manuscript.
Comment 9:  Fig. 1F: Higher resolution images would help the readers see how similar/different the two virus preparations are in their infectivity.

Response 9:  We have replaced the images with those of higher resolution and modified the arrangement of the photographs in the revised manuscript.

Comment 10: Line 20, 74, 127, 132, 140, etc: Please be consistent with whether you use sIgA or SIgA.

Response 10:  I have standardized the format of SIgA throughout the entire manuscript to SIgA.
Comment 11: Fig. 2 and Fig. 3: There is no mention of the size of groups used for the animal studies, either in the methods, figures, or figure legends. For Fig.2B-E and Fig.3B-C represent individual animals?  Please clarify this.

Response 11:   We have added the number of animals in each treatment group to the figure captions for the animal experiments in the revised manuscript.

Fig2 (A). Schematic diagram of the experimental design. Briefly, groups of 6~8-week-old A129 mice (n=6) were immunized with 104 PFU of YF-17D or aerosolized YF-17D, respectively. Then, sera samples at 14, 21 dpi were collected for total YFV-specific IgG antibody.

Fig2 (E). SIgA antibody in BAL of YF-17D-immunized mice (n=5) at 21 dpi were measured by ELISA.

Fig3 (A). ELISpot results measured with the AID ELISpot reader. To detect YF-17D-specific T-cell responses, cytokine screening of splenocytes was performed by IFN-γ and IL-4 ELISPOT, splenocytes harvested from A129 immunized (n=3~4) with 1 × 104 PFU of YF-17D at three weeks post-vaccination.

Fig4 (A). Protective efficacy of aerosolized YF-17D in A129 mice. (A) Survival curve of the immunized mice (n=5).

Fig4 (C). Tissue distribution in mice (n=3) at 5 day after YF-17D challenge. Dotted lines represent limits of detection. The data were shown as mean± SD. Significance was calculated using one way ANOVA (n.s., not significant; *, P < 0.05; ***, P < 0.001; ****, P<0.0001).
Comment 12: Line 321: “17D-special” should be “17D-specific”

Response 12: We have changed “17D-special” to “17D-specific” at line 425 in the revised manuscript.

Round 2

Reviewer 2 Report

Comments and Suggestions for Authors

The authors have sufficiently addressed the concerns raised.

Comments on the Quality of English Language

Edits made have improved the language concerns raised.